# MixDiffusion: Mixing Diffusion-based Uni-condition Text-to-Image Generation Models for Multi-conditions Image Synthesis

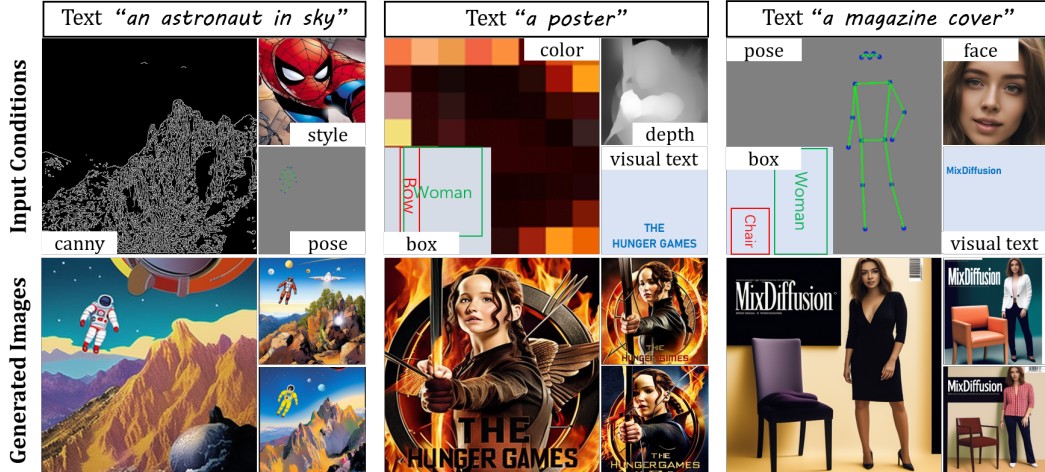

Figure 1: **Images Generated with our proposed MixDiffusion.** Each column shows a controllable image generation example. The upper part of each column shows the provided control conditions, and the lower part presents the generated images by the proposed MixDiffusion under these control conditions. MixDiffusion is training-free, and can support an unlimited number of control condition inputs.

## ABSTRACT

Recently, there has been a notable surge in the field of text-to-image (T2I) generation thanks to the powerful generation ability of diffusion models. To achieve controllable image generation, researchers explore to leverage additional control conditions besides text to guide image generation with diffusion models. Despite their advancements, most existing diffusion-based T2I generation models are primarily focused on one type of condition (uni-condition) controllable image generation. However, to better cater to users' increasing demands for more detailed and creative outputs, there is a growing need for models to simultaneously support multiple control conditions (multi-conditions) for controllable image generation. In this work, we seek to expand the control conditions for diffusion-based T2I generation models and propose MixDiffusion, a training-free diffusion-based image generation framework supporting an unlimited number of control conditions, e.g., bounding boxes, keypoints, sketches, etc., besides text by incorporating multiple pre-trained diffusion based uni-condition image generation models to collaboratively synthesize images that satisfy all specified control conditions. The key insight of the proposed approach is to derive the predicted noise distribution in each denoising step of the diffusion-based multi-conditions image generation model MixDiffusion from the predicted noise distributions of multiple diffusion-based uni-condition models with a derived integration formula, which is supported by rigorous theory proof. The training-free manner makes the proposed MixDiffusion highly convenient for deployment and for adding new type of control conditions as needed.

# 1 INTRODUCTION

How can one accurately convey the images in his mind to another person? As the saying goes, "a picture is worth a thousand words". Drawing is undeniably a more intuitive and precise way than textual description. Thanks to the significant advancements in the T2I synthesis community Goodfellow et al. (2020); Ho et al. (2020); Song et al. (2020), it is now possible to generate realistic and high-fidelity images within seconds using advanced models. T2I synthesis models, such as Stable Diffusion Rombach et al. (2022); Podell et al. (2023), Imagen Saharia et al. (2022), and DALL-E Ramesh et al. (2022); Betker et al. (2023), have shown remarkable capabilities in creating high-quality images with impressive aesthetic and realistic features. However, these models often struggle with generating images with controllable details, e.g., producing particular visual text, placing objects in precise locations, or rendering people in specific poses.

In response to these limitations, substantial progress has been made in controllable image generation. By providing diffusion models with additional inputs beyond textual descriptions, e.g., bounding boxes in BoxDiff Xie et al. (2023) and InstanceDiffusion Wang et al. (2024), keypoints in the GFLA framework Ren et al. (2020) and PIDM Bhunia et al. (2023), and segmentation masks in ALDM Li et al. (2024) and SceneComposer Zeng et al. (2022), the controllable image generation models have been able to generate images following specific control condition inputs. Since these models support only one customized control condition besides text description to guide the generation, we name them as 'uni-condition image generation models'.

Despite achieving notable success through either fine-tuning or training from scratch, one of the biggest limitations of the uni-condition image generation models is that only one type of condition is supported to guide the image generation. For example, if we aim to generate a fashion magazine cover featuring a charming girl with particular pose using DreamPose Karras et al. (2023), which accepts keypoints as an input control condition, the resulting image would likely satisfy the desired pose. However, it is very challenging or even impossible for DreamPose to generate a girl with a specific face (e.g., a face in a reference image) if the generated girl is not a celebrity. Recently, a few diffusion-based image generation models, e.g., Uni-ControlNet Zhao et al. (2024) and Diffblender Kim et al. (2023), are proposed, which can support more than one control conditions besides text description. Unfortunately, these models can only accept a very limited number of control conditions, and the control conditions are fixed for each model. Retraining the model to accommodate additional control conditions is both time-consuming and expensive. Moreover, as user demands evolve, the range of possible conditions for controllable image generation expands, resulting in countless potential combinations.

To overcome the above limitations of the existing controllable image generation models, we introduce MixDiffusion, a training-free framework that supports an unlimited number of control condition inputs by mixing multiple pre-trained diffusion based uni-condition image generation models to collaborate in producing a single image. Moreover, to better mix these models to generate images with higher quality, we derive an integration strategy for the proposed MixDiffusion, which is more reasonable and theoretically interpretable, and boosts MixDiffusion to achieve superior generation results when combining multiple models. As shown in Fig.1, the images generated by MixDiffusion perfectly satisfy all control condition inputs.

The main contributions of this work are as follows:

- We propose a training-free framework for integrating multiple diffusion based uni-condition T2I generation models, supporting flexible and an unlimited number of control conditions to intricately control the image generation. This fills the gap between real-world applications requiring multi-conditions controllable image generation and the current T2I generation models that predominantly support only one or two control conditions. Additionally, the training-free property of the proposed framework is highly convenient for deployment and for adding new type of control conditions as needed.

- We extend the theory system of diffusion models by deriving the predicted noise distribution of a diffusion based multi-conditions image generation model from the predicted noise distributions of multiple diffusion based uni-condition models, and provide the rigorous theory proof. This offers a novel perspective for multi-conditions image synthesis task.

- Extensive experiments demonstrate the flexibility of our proposed framework in integrating multiple diffusion-based T2I generation models, as well as its superior performance on controllable and high-fidelity image generation.

## 2 RELATED WORK

### 2.1 CONTROLLABLE IMAGE SYNTHESIS

Layout, as a versatile and user-friendly conditional input, has been extensively explored in the field of controllable image generation. Layouts are typically categorized into bounding boxes, segmentation masks Li et al. (2024); Zeng et al. (2022), scribbles Carrillo et al. (2023); Zhang et al. (2023), and more. InstanceDiffusion Wang et al. (2024) has achieved notable success in layout-to-image synthesis by converting various layouts into point representations, allowing a single model to accept multiple types of condition inputs. However, at its core, it still only processes point-based conditions.

Some approaches transform layouts into textual representations and concatenate them with textual descriptions Yang et al. (2023), but incorporating different layout combinations or adding additional conditions still necessitates retraining the model, which is both time-consuming and costly. In certain cases, finer control is required, such as specifying a person's pose through keypoints for hands or body Narasimhaswamy et al. (2024); Li et al. (2023), or even 3D meshes Ge et al. (2019); Zhang et al. (2024b). Additionally, image synthesis models often struggle with generating visual text, i.e., signs and banners in the generated images often contain spelling mistakes or incorrectly rendered letters. To address this, some studies have incorporated text layouts into diffusion models, enabling accurate visual text generation in the image synthesis task Chen et al. (2024; 2023).

Moreover, other works have tackled image-guided image generation Tumanyan et al. (2023), created images in various styles (such as oil painting and cartoon styles) Sun et al. (2023), and generated images with specific faces Ye et al. (2023a). These advancements have greatly enriched the field of controllable T2I synthesis. As a result, our work, a training-free framework for integrating diffusion-based image generation models, has become increasingly relevant and necessary for supporting more flexible and diverse control conditions in image generation.

### 2.2 MULTI-CONDITIONS IMAGE SYNTHESIS

The growing interest in controllable image generation has led to the development of models such as Collaborative Diffusion Huang et al. (2023), Diffblender Kim et al. (2023), Uni-ControlNet Zhao et al. (2024) and T2I-Adapter Ye et al. (2023b), which can accept more than one condition inputs to guide the image synthesis. However, these models face limitation due to the difficulty in collecting and annotating diverse training datasets, as well as the high training costs. Consequently, they often support only two or three fixed control condition inputs, and modifying input combinations or adding new type of control conditions typically requires retraining the model.

In contrast, MixDiffusion overcomes this limitation by mixing existing pre-trained diffusion-based image generation models to flexibly support various control conditions for controllable image generation in a training-free manner. This allows greater adaptability and efficiency for finer controllable image generation with multiple condition inputs.

## 3 PRELIMINARIES: STABLE DIFFUSION

Stable Diffusion Rombach et al. (2022) consists of two primary components: an autoencoder consisting of an encoder $\mathcal{E}$ and a decoder $\mathcal{D}$, and a denoiser $\epsilon_\theta$. The encoder $\mathcal{E}$ maps images into a latent feature space, i.e., $z_0 = \mathcal{E}(x_0)$ where $z_0$ is the encoded image feature and $x_0$ represents the original image, reducing feature dimensionality to save computational memory and speed up inference. The decoder $\mathcal{D}$ is responsible for mapping the denoised feature back into the image space, i.e., $x_0 = \mathcal{D}(z_0)$.

The diffusion process is divided into a *forward process* and a *reverse process*. The forward process, which is used in training, incrementally adds small amounts of standard Gaussian noise $\epsilon$ to a clean

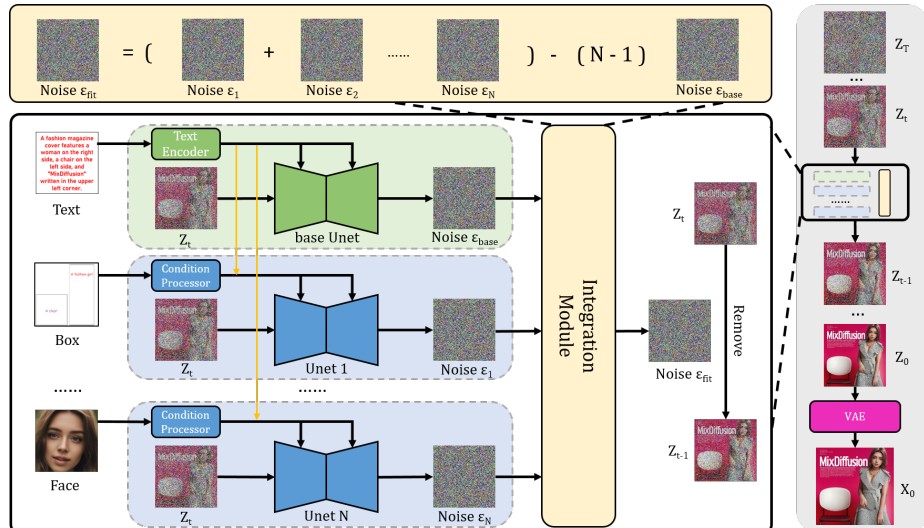

Figure 2: The overall architecture of MixDiffusion. Inputs from various modalities are processed by the Processor to extract features. Then pre-trained uni-condition Unet accept condition features to predict the noise. Finally the predicted noises are integrated by Integration Module to get the noise to be removed.

latent image $z_0$, eventually transforming it into standard Gaussian noise $z_T$. In contrast, in the reverse process, Gaussian noise $z_T$ is sampled and progressively denoised over $T$ steps, transforming the noise $z_T$ into a clean latent image $z_0$. At each step, the denoiser $\epsilon_\theta$ predicts the noise $\epsilon_t$ to be removed,

$$\epsilon_t = \epsilon_\theta(z_t, t) \tag{1}$$

$$z_{t-1} = \frac{z_t - (1 - \sqrt{1 - \overline{\alpha_t}})\epsilon_t}{\sqrt{\overline{\alpha_t}}}, \tag{2}$$

where $\{\overline{\alpha_t}\}_{t=1}^T$ are hyperparameters, $t \in T$ represents the current time step, and $\theta$ denotes the model parameters. The model $\epsilon_\theta$ can then be trained via:

$$\mathcal{L} = \mathbb{E}_{z \sim \mathcal{E}(x), \epsilon \sim \mathcal{N}(0,1), t}[||\epsilon - \epsilon_\theta(z_t, t)||_2^2], \tag{3}$$

where $t \in T$ is the training time step.

## 4 METHOD

In this section, we first introduce the framework of the proposed MixDiffusion in Sec.4.1. Then, the integration formula for MixDiffusion is derived in Sec.4.2. Finally, we introduce an adjustment strategy of the integration intensity in Sec.4.3, aiming at further improving the quality of the generated image and the consistency between generated image and control conditions.

### 4.1 ARCHITECTURE OF MIXDIFFUSION

As shown in Fig.2, We divide a controllable image generation model into four key components: a Condition Processor for preprocessing the control conditions, a Text Encoder, a denoiser (i.e., Unet), and a Variational Autoencoder (VAE). Usually, each uni-condition image generation model comes with its own pre-trained VAE and Text Encoder. It will be quite memory-consuming and computation-costing to use separate VAE and Text Encoder for each uni-condition model. Fortunately, we find that when all models share the same VAE and Text Encoder, they can maintain strong performance and generate high-quality images. Therefore, the same VAE and Text Encoder are adopted for all these models, which significantly streamlines the integration process. In each denoising step, each control condition is first processed by the pre-trained feature process to obtain the condition feature, and then input into the pre-trained denoiser (i.e., Unet) to predict a noise.

These noises are integrated by the Integration Module to get the final integrated noise, which will be removed in this denoising step. After $T$ iterations of denoising, the final latent is obtained. Subsequently, this latent is decoded by the VAE to produce the final generated image.

## 4.2 INTEGRATION MODULE

Each denoising step of the reverse diffusion process can be interpreted as sampling a noise from a Gaussian distribution:

$$\epsilon_{t-1} \sim \mathcal{N}(u_t(z_t, t), \sigma_t^2), \tag{4}$$

where $u_t$ represents the mean of the Gaussian distribution at step $t$, predicted by the denoiser $\epsilon_\theta$ based on the latent variable $z_t$ and the current step $t$. $\sigma_t$ denotes the standard deviation of the predicted Gaussian distribution. In this work, we employ the DDIM scheduler Song et al. (2020), where the values $\{\sigma_t\}_{t=1}^T$ are constants, and the denoiser only predicts $u_t$. This simplification enhances the inference process and contributes to a more elegant final integration formula.

The core idea of MixDiffusion's Integration Module is to treat the predicted noises of uni-condition controllable image generation models as distributions $\{\mathcal{N}(\epsilon_{t-1}; u_{t,i}(z_t, C_i), \sigma_t^2)\}_{i=1}^N$, where $C_i$ represents the condition input and $N$ indicates the number of integrated uni-condition models. These distributions are utilized to calculate the predicted noise of the integrated model, which can be treated as a virtually trained multi-conditions image generation model accepting multiple condition inputs, $\mathcal{N}(\epsilon_{t-1}; u_t(z_t, C_1, C_2, \ldots, C_N), \sigma_t^2)$. By employing Bayes' Rule, we can derive the integration formula:

$$P(\epsilon_{t-1}|C_1, \ldots, C_N, z_t) = \frac{\prod_{i=1}^N P(\epsilon_{t-1}|C_i, z_t)}{\prod_{i=1}^{N-1} P(\epsilon_{t-1}|z_t)}, \tag{5}$$

where $P(\epsilon_{t-1}|C_i, z_t), i \in N$ denotes the distribution of the uni-condition controllable diffusion models, while $P(\epsilon_{t-1}|z_t)$ corresponds to a base diffusion model that does not accept additional conditions. Here, we assume that the classifiers of each model are independent of each other, i.e., $P(C_1, C_2, \ldots C_N|\epsilon_{t-1}, z_t) = \prod_{i=1}^N P(C_i|\epsilon_{t-1}, z_t)$. Intuitively, this assumption aligns with human cognition, since each diffusion-based uni-condition model is trained independently.

We select the sample with the highest probability density as the sampling noise of the integrated model, leading to a concise and elegant integration formula:

$$\begin{cases} \epsilon_{\text{fit}} = \arg\max_\epsilon \dfrac{\prod_{i=1}^N P(\epsilon|C_i, z_t)}{\prod_{i=1}^{N-1} P(\epsilon|z_t)} \\ P(\epsilon|C_i, z_t) = \dfrac{1}{\sqrt{2\pi}\sigma} \exp\left(-\dfrac{(\epsilon - \epsilon_i)^2}{2\sigma^2}\right) \Rightarrow \\ P(\epsilon|z_t) = \dfrac{1}{\sqrt{2\pi}\sigma} \exp\left(-\dfrac{(\epsilon - \epsilon_{\text{base}})^2}{2\sigma^2}\right) \end{cases} \quad \begin{aligned} & \epsilon_{\text{fit}} = \arg\min_\epsilon \left(\sum_{i=1}^N (\epsilon - \epsilon_i)^2 - (N-1)(\epsilon - \epsilon_{\text{base}})^2\right) \\ & \epsilon_{\text{fit}} = \sum_{i=1}^N \epsilon_i - (N-1)\epsilon_{\text{base}} \end{aligned} \tag{6}$$

where $\epsilon_{fit}$ denotes the predicted noise of the integrated multi-conditions image generation model that supports multiple control conditions, $\epsilon_i$ represents the predicted noise of each individual uni-condition controllable diffusion model, and $\epsilon_{base}$ indicates the predicted noise of a diffusion model that does not accept additional condition inputs.

## 4.3 INTEGRATION INTENSITY STRATEGY

To further improve the image quality and the consistency between control conditions and image content, we propose to use a dynamic integration intensity for each denoising step. This strategy is effective for two primary reasons. First, an important assumption of Eq.(6) is that the classifiers of different diffusion models are independent of each another. However, this assumption holds reasonably well during the early denoising steps but becomes less reliable in the later steps and usually adding high intensity condition in later denoising steps degrades the image quality Zhang et al.

| settings | OKS | | IoU | |
|---|---|---|---|---|
| | AP@0.5 ↑ | AP@0.75 ↑ | AP@0.5 ↑ | AP@0.75 ↑ |
| Uni-ControlNet | 65.67% | 33.96% | - | - |
| Diffblender | 45.37% | 17.69% | 70.01% | 48.43% |
| MixDiffusion | **88.09%** | **47.30%** | **88.51%** | **73.36%** |

Table 1: A quantitative comparison of Uni-ControlNet, Diffblender, and MixDiffusion (ours) on the COCO2017 validation set across five control conditions: Text, Box/Pose, Canny, Depth, and Sketch. It should be noted that Uni-ControlNet does not support bounding box control.

| Methods | FID↓ | CLIP↑ | Depth(RMSE)↓ | Canny(RMSE)↓ | OKS(AP@0.5)↑ |
|---|---|---|---|---|---|
| CnC | 42.18 | 14.10 | 35.42 | - | - |
| Cocktail | 45.12 | 14.41 | 32.78 | - | 75.82% |
| T2I-Adapter | 46.28 | 14.27 | **30.51** | 91.78 | - |
| AnyControl | **38.93** | 14.31 | 32.97 | 105.65 | 59.50% |
| DynamicControl | 42.46 | 13.63 | 33.92 | 103.89 | 76.55 |
| MixDiffusion | 39.81 | **14.42** | 31.21(43.87) | **90.32** | **86.03%** |

Table 2: Comparison of multi-control methods on COCO2017 with four control conditions (Text, Depth, Canny and Pose). In our experimental setup, the original MixDiffusion framework using Gligen achieved a depth RMSE of 43.87. By replacing GLIGEN with T2I-Adapter in this specific experiment, we observed significant improvement with RMSE reduced to 31.21.

(2023); Li et al. (2023). Thus, decreasing the integration intensity in the later steps can help mitigate the negative effects associated with the breakdown of this assumption. Second, as we known, during the $T$ denoising steps of the diffusion model, the early steps primarily determine the object's contours and positions, while the later steps focus on adding details to the image Guo et al. (2025). Consequently, in our integration strategy, we gradually decrease the integration strength, allowing the base Unet to serve as the primary denoiser in the later steps to depict more content details. This integration intensity adjustment strategy ensures that the image details are less susceptible to errors, resulting in a more coherent generated image. The mathematical formula for adjusting the integration intensity is as follows:

$$\epsilon_{t-1} = w\epsilon_{fit} + (1-w)\epsilon_{base}, \qquad (7)$$

where $w$ represents the integration strength, which is modeled as a quadratic function of the steps,

$$w = (t/T)^2 \in [0, 1]. \qquad (8)$$

## 5 EXPERIMENT

In this section, we outline the experimental setup in Sec.5.1, including models, hyper-parameter settings, datasets, and evaluation metrics. In Sec.5.2, the proposed MixDiffusion is compared with state-of-the-art multi-conditions image generation models. Then ablation study is conducted in Sec.5.3 to analyze the contribution of each proposed module/strategy to MixDiffusion. Finally, in Sec.5.4, we present quantitative results that compare the consistency of generated images with text descriptions and assess image quality after integrating various numbers of models.

### 5.1 EXPERIMENTS SETUP

Theoretically, the proposed MixDiffusion can mix unlimited diffusion-based uni-condition T2I generation models, which means that MixDiffusion can take unlimited control conditions as input to control the image generation, as long as there are pre-trained diffusion-based uni-condition image generation models supporting these control conditions. In this work, we use six pre-trained diffusion-based image generation models to illustrate the performance of MixDiffusion, TextDiffuser V2 Chen et al. (2023), T2I-Adapter Mou et al. (2024), Gligen Li et al. (2023), IP-Adapter

| settings | IoU | |
|---|---|---|
| | AP@0.5 ↑ | AP@0.75 ↑ |
| TBD-I | 67.17% | 56.27% |
| TBD-S | 53.96% | 38.03% |
| TBD | **86.57%** | **72.00%** |
| TBDS-I | 66.44% | 55.56% |
| TBDS-S | 59.27% | 45.45% |
| TBDS | **87.79%** | **72.57%** |
| TBDSC-I | 65.42% | 55.28% |
| TBDSC-S | 63.02% | 50.90% |
| TBDSC | **88.51%** | **73.36%** |

Table 3: Ablation study on the COCO2017 dataset. The suffix '-I' indicates the ablation of Integration Intensity in MixDiffusion, while the suffix '-S' indicates replacing the proposed integration strategy with interpolation.

| settings | AestheticScore ↑ | ImageReward | |
|---|---|---|---|
| | | rating↑ | artifact↓ |
| TB | 4.46 | 4.30 | 2.92 |
| TBD | 4.78 | 4.47 | 2.88 |
| TBDS | 4.92 | 4.73 | 2.82 |
| TBDSC | **5.15** | **4.73** | **2.80** |
| TP | 4.70 | 3.73 | 3.24 |
| TPD | 4.75 | 3.97 | 3.19 |
| TPDS | 4.97 | 4.15 | 3.14 |
| TPDSC | **5.33** | **4.19** | **3.13** |

Table 4: Evaluation about image quality on the COCO2017 Validation Set. TBPCDS stands for Text, Bounding Box, Pose, Canny Edge, Depth Map and Sketch.

Ye et al. (2023b), DreamShaper Lykon (2023), and Stable Diffusion V1.5 Rombach et al. (2022), which support different control conditions beyond textual descriptions, including bounding boxes, poses, faces, visual texts, segmentation maps, etc. Text encoder and VAE are shared by these six models to reduce computational memory and time. When simultaneously loading four condition processors along with their corresponding denoisers, the GPU memory usage reaches 16 GB, and a single image can be generated in approximately 8.8 seconds.

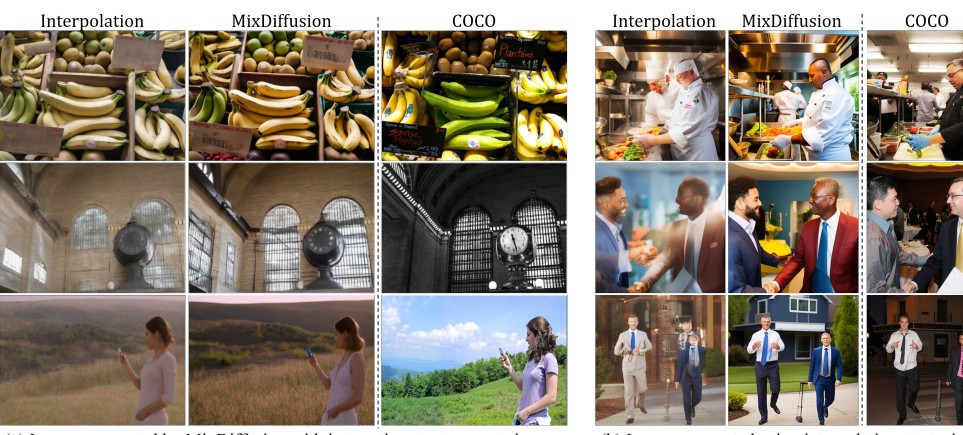

(a) Images generated by MixDiffusion with integration strategy contain a greater amount of high-frequency information.

(b) Images generated using interpolation strategies are prone to inducing illusions.

Figure 3: Comparison of images generated using the interpolation and the integration strategy. Experiments conducted on the COCO2017 Evaluation Set. 'COCO' stands for original reference images in the dataset.

Due to its training-free nature, the proposed MixDiffusion does not require a training stage. We directly evaluate its performance using subsets of the COCO2017Eval Lin et al. (2014) dataset, which contains images of people, and the MARIOEval Zhang et al. (2024a) dataset, considering various evaluation perspectives. To assess consistency with human preferences and the realism of generated images, we adopt the ImageReward score Xu et al. (2023) and the Fréchet Inception Distance (FID). A higher ImageReward score indicates better alignment with human preferences, while a lower FID score reflects higher visual realism. We also use a pretrained model discus0434 & Goswami (2025) to evaluate the aesthetic quality of the generated images.

To measure alignment between the generated images and the input textual descriptions, we use the CLIP-score Zhengwentai (2023), where higher scores signify better semantic alignment. To

evaluate how well the generated images adhere to control conditions, we use different metrics: for bounding boxes, we compute the Intersection over Union (IoU) between the predicted and ground-truth boxes; for keypoints, we apply Object Keypoint Similarity (OKS). For structural conditions such as Canny edges and depth maps, we use the Root Mean Square Error (RMSE) to quantify pixel-level deviations between the generated results and the control inputs—lower RMSE values indicate better preservation of edge details and depth information.

## 5.2 COMPARISON TO PREVIOUS METHODS

Since there is no training-free multi-conditions image generation model, we compare the proposed MixDiffusion with several state-of-the-art methods for multi-conditions image generation, Uni-ControlNet Zhao et al. (2024) , Diffblender Kim et al. (2023), AnyControl Sun et al. (2024), Cocktail Hu et al. (2023), DynamicControl He et al. (2025) and CnC Lee et al. (2024), which all require training with different combinations of control inputs. We detect the keypoints, bounding boxes, canny edge and depth map in the generated images with off-the-shelf detection algorithm. Then measure the similarity between the detected keypoints, bounding boxes, canny edge and depth map between generated images and the ground truth with OKS, IoU and RMSE metrics. The quantitative results are summarized in Table 1 and Table 2, which shows that our proposed MixDiffusion surpass other models by a large margin, indicating that MixDiffusion can generate images that follow the control conditions much better than the other seven models.

## 5.3 ABLATION STUDY

In this section, we replace MixDiffusion's integration strategy with interpolation and remove the integration intensity to respectively evaluate the contributions of these two modules to the overall performance of MixDiffusion.

### 5.3.1 INTEGRATION STRATEGY

Instead of using the proposed integration strategy to mix uni-condition image generation models in MixDiffusion, a more straightforward way is to interpolate these models, i.e., taking the average of the predicted noises of each uni-condition models as the final predicted noise,

$$\epsilon_{t-1} = \frac{\sum_{i=1}^{N} \epsilon_i}{N}. \tag{9}$$

To demonstrate the effectiveness of the proposed integration strategy, we integrate five uni-condition diffusion models with the proposed integration strategy and the interpolation strategy, respectively. The five models receive conditional inputs of text (T), bounding boxes (B), canny edge (C), depth (D), and sketch (S) respectively. The quantitative comparison results are shown in Table 3, which indicates that when integrating a larger number of models, the interpolation method fails to align the generated images closely with control conditions. This can be explained through the interpolation formula, which suggests that as more models are integrated, the contribution of each component diminishes, weakening the overall control strength. In contrast, the integration strategy Eq.(6) does not encounter this issue, as the contribution of each control model remains constant regardless of the number of models. Moreover, the images generated by the interpolation method are with lower quality, as depicted in Fig.3, which loss significant high-frequency information and introduce faint artifacts due to the direct summation of predictions from different models.

### 5.3.2 INTEGRATION INTENSITY

Table 3 also compares the image generation results of MixDiffusion with and without the integration intensity adjustment strategy (i.e., TBC-I V.S. TBC, TBCD-I V.S. TBCD, TBCDS-I V.S. TBCDS). in terms of IoU. The results show that the model consistently performs better when intensity control is applied. MixDiffusion with integration intensity adjustment produces images with a more harmonious integration of components and better alignment with the conditional inputs compared to MixDiffusion without integration intensity adjustment. This improvement stems from an important phenomenon: in the later denoising steps of generation, the predictions of individual models diverge, and their combination can introduce competition, resulting in less cohesive outputs.

| settings | OKS | | IoU | |
|---|---|---|---|---|
| | AP@0.5 ↑ | AP@0.75 ↑ | AP@0.5 ↑ | AP@0.75 ↑ |
| B/P+T | 85.10% | 45.94% | 84.55% | 69.86% |
| B/P+TD | 85.98% | 46.96% | 86.57% | 72.00% |
| B/P+TDS | 86.03% | **47.87%** | 87.79% | 72.57% |
| B/P+TDSC | **88.09%** | 47.30% | **88.51%** | **73.36%** |
| COCO | 91.22% | 83.31% | 75.45% | 49.32% |

Table 5: Evaluation on the COCO2017 Validation Set. Primary comparisons focus on B/P (Box, evaluated using IoU, and Pose, assessed via OKS), with additional conditions TDSC (Text, Depth, Sketch, and Canny edge).

## 5.4 FURTHER ANALYSIS OF THE INTEGRATION MODULE

To further demonstrate the effectiveness of the proposed integration strategy, we conduct experiments to generate image under tow conditions (text and bounding box, B+T / text and pose, P+T), three conditions (adding depth, B/P+TD), four conditions (adding sketch, B/P+TDS), and five conditions (adding canny edge, B/P+TDSC). As shown in Table 5 and Table 4, the OKS, IoU, aesthetic scores and ImageReward improve with increasing control conditions.

To simulate a scenario where the information provided by the control conditions is entirely independent, we designed experiments incorporating control conditions based on text (T), visual text (Vt), color (Co), and style (Sy). In these conditions, the visual text input in the local control condition is independent of the other three global conditions. By observing the accuracy of visual text generation, we can evaluate performance when the control conditions are entirely independent, in comparison to scenarios with strong coupling, such as canny and sketch. As shown in Table 6, the image quality metrics, including CLIP score and ImageReward, improved, while the accuracy of generating visualized text experienced a slight but acceptable decline.

| settings | CLIP-Score↑ | ImageReward | | OCR | |
|---|---|---|---|---|---|
| | | rating↑ | artifact↓ | ACC.T↑ | ACC.B@0.5↑ |
| TVt | 35.67 | 3.87 | 3.23 | 76.90 | 95.93 |
| TVtCo | 35.98 | 4.09 | 3.18 | 76.55 | 95.58 |
| TVtCoSy | **36.03** | **4.08** | **3.19** | 76.53 | 95.48 |

Table 6: Evaluation results of incorporating additional unique control conditions in MixDiffusion. ACC.T represents the accuracy of the text to be rendered, while ACC.B@0.5 refers to the IoU@0.5 for the location of the visualized text.

## 6 CONCLUSION

Currently, there is a significant gap between increasing demand for fine-grained control in image generation and the capabilities of the existing controllable image generation models. To fill this gap, we propose MixDiffusion, a novel training-free controllable image generation framework supporting various control conditions by mixing multiple off-the-shelf pre-trained diffusion-based uni-condition image generation models. An integration formula is derived with theory proof to better integrate multiple diffusion-based image generation models. We believe the integration formula extends the theory system of the diffusion model. The training-free manner increases the flexibility of the proposed MixDiffusion to include additional new type of control conditions as needed. Extensive experiments are conducted to demonstrate the effectiveness of the proposed MixDiffusion, which shows superiority in terms of both image quality and alignment with input control conditions of the generated images.

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

# A  APPENDIX

## A.1  PROOF OF THE INTEGRATION FORMULA

In this section, we provide a detailed derivation of the integration formula. We first establish the relationship between the Gaussian distributions of the uni-condition image generation models and the one of the multi-conditions image generation model. Following that, we derive the integrated noise for the multi-conditions image generation model, which is removed in each denoising step, from the individual noises of the uni-condition image generation models.

$$
\begin{aligned}
P(\epsilon_{t-1}|C_1, C_2, &\ldots C_N, z_t) \\
&= \frac{P(C_1, C_2, \ldots C_N, \epsilon_{t-1}|z_t)}{P(C_1, C_2, \ldots C_N|z_t)} \\
&= \frac{P(C_1, C_2, \ldots C_N|\epsilon_{t-1}, z_t)P(\epsilon_{t-1}|z_t)}{P(C_1, C_2, \ldots C_N|z_t)} \\
&\overset{①}{=} \frac{(\prod\limits_{i=1}^{N} P(C_i|\epsilon_{t-1}, z_t))P(\epsilon_{t-1}|z_t)}{\prod\limits_{i=1}^{N} P(C_i|z_t)} \\
&= \frac{(\prod\limits_{i=1}^{N} \frac{P(\epsilon_{t-1}|C_i, z_t)P(C_i|z_t)}{P(\epsilon_{t-1}|z_t)})P(\epsilon_{t-1}|z_t)}{\prod\limits_{i=1}^{N} P(C_i|z_t)} \\
&= \frac{\prod\limits_{i=1}^{N} P(\epsilon_{t-1}|C_i, z_t)}{\prod\limits_{i=1}^{N-1} P(\epsilon_{t-1}|z_t)}
\end{aligned}
\tag{10}
$$

In ①, we assume that the classifiers of each model are independent of each other, i.e., $P(C_1, C_2, \ldots C_N|\epsilon_{t-1}, z_t) = \prod\limits_{i=1}^{N} P(C_i|\epsilon_{t-1}, z_t)$. Intuitively, this assumption aligns with human cognition, since each diffusion-based uni-condition model is trained independently.

To determine the sampling result of the multi-conditions image generation model, we select the sample with the highest probability density. This corresponds to identifying the noise $\epsilon$ that maximizes the conditional probability $P(\epsilon_{t-1}|C_1, C_2, \ldots C_N, z_t)$. In other words, we find the noise to be removed by maximizing the Eq.(10).

$$\epsilon_{fit} = \arg\max_{\epsilon} \frac{\prod_{i=1}^{N} P(\epsilon|C_i, z_t)}{\prod_{i=1}^{N-1} P(\epsilon|z_t)}$$

$$\overset{\textcircled{2}}{=} \arg\max_{\epsilon} \frac{\prod_{i=1}^{N} \frac{1}{\sqrt{2\pi}\sigma} \exp\left(-\frac{(\epsilon-\epsilon_i)^2}{2\sigma^2}\right)}{\prod_{i=1}^{N-1} \frac{1}{\sqrt{2\pi}\sigma} \exp\left(-\frac{(\epsilon-\epsilon_{base})^2}{2\sigma^2}\right)}$$

$$= \arg\max_{\epsilon} \frac{1}{\sqrt{2\pi}\sigma} \exp\left(-\frac{\left(\sum_{i=1}^{N}(\epsilon-\epsilon_i)^2\right) - (N-1)(\epsilon-\epsilon_{base})^2}{2\sigma^2}\right)$$

$$= \arg\min_{\epsilon} \sum_{i=1}^{N}(\epsilon-\epsilon_i)^2 - (N-1)(\epsilon-\epsilon_{base})^2$$

$$= \sum_{i=1}^{N} \epsilon_i - (N-1)\epsilon_{base}$$

(11)

In ②, we set the standard deviation $\sigma$ of the Gaussian distributions to a constant value. This is because, in DDPM, $\sigma$ could serves as a hyperparameter, and the model only predicts the mean $u$. In other noise schedulers, if the standard deviation varies, the final integration formula will include a term involving $\sigma$ within $\epsilon_i$.

## A.2 EXPERIMENT SETTINGS

To make the project reproducable, we introduce the experiment settings in more details.

When calculating COCO OKS, the standard error K for each keypoint is treated as an empirical constants, with our settings as follows: K = [0.26, 0.25, 0.25, 0.35, 0.35, 0.79, 0.79, 0.72, 0.72, 0.62, 0.62, 1.07, 1.07, 0.87, 0.87, 0.89, 0.89] / 10.0

It is worth noting that Uni-ControlNet does not support bounding box condition inputs. Therefore, in the comparison experiments in which bounding box is included as control condition input, the generation results of the Uni-ControlNet is not presented (e.g., Fig.4 in this supplementary material). For the individual Uni-condition image models used in the MixDiffusion, we use the recommended hyperparameters and configurations provided in their papers or projects.

## A.3 ADDITIONAL COMPARISON RESULT

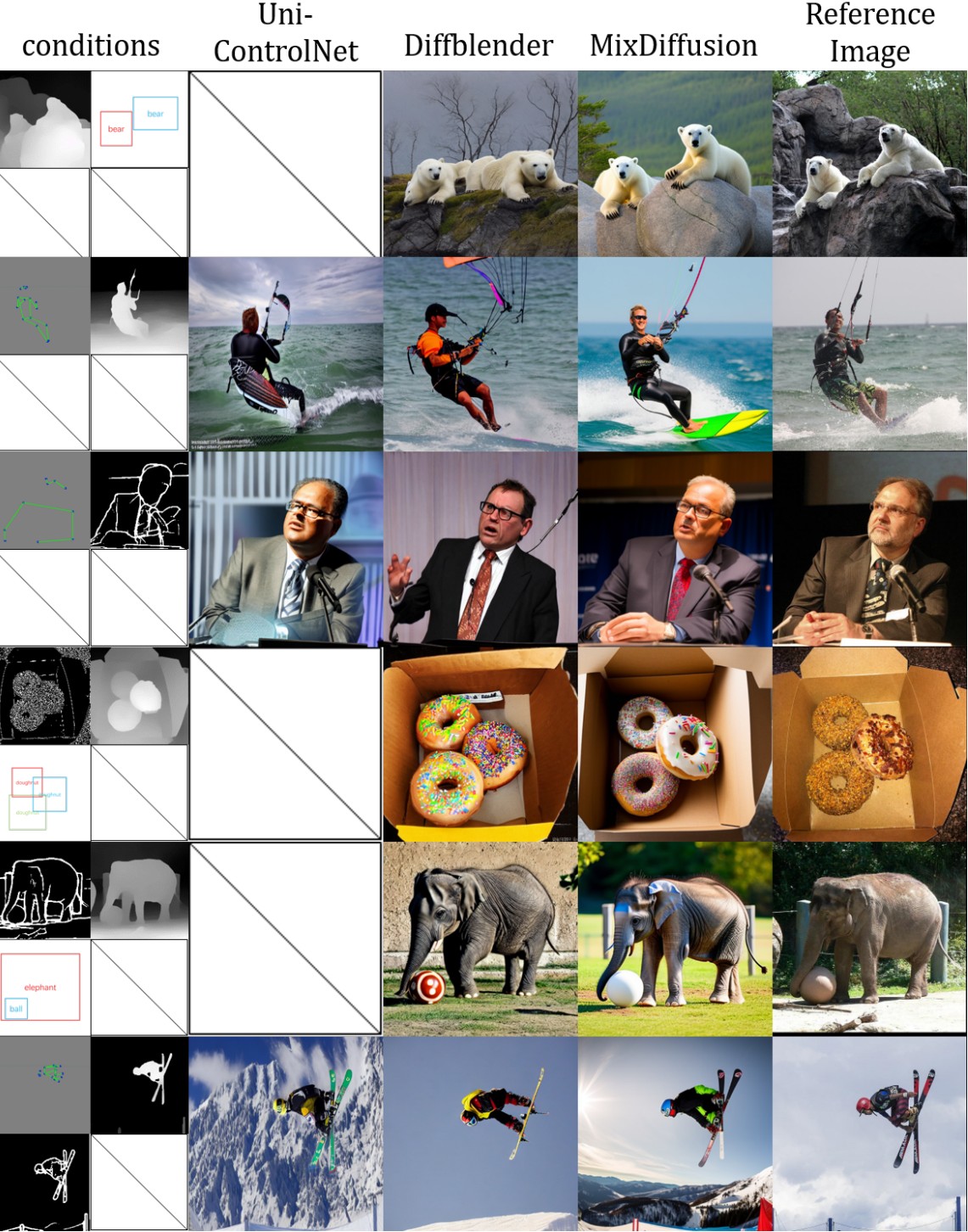

Figure 4: Comparison results of different controllable image generation models. Generation results of the Uni-ControlNet are not presented in the case of bounding box is included in the control condition inputs (i.e., the first rows) since the Uni-ControlNet does not support bounding box input. Uni-ControlNet and DiffBlender perform worse than MixDiffusion in terms of background generation, consistency between generated images and conditions, and image detail quality.

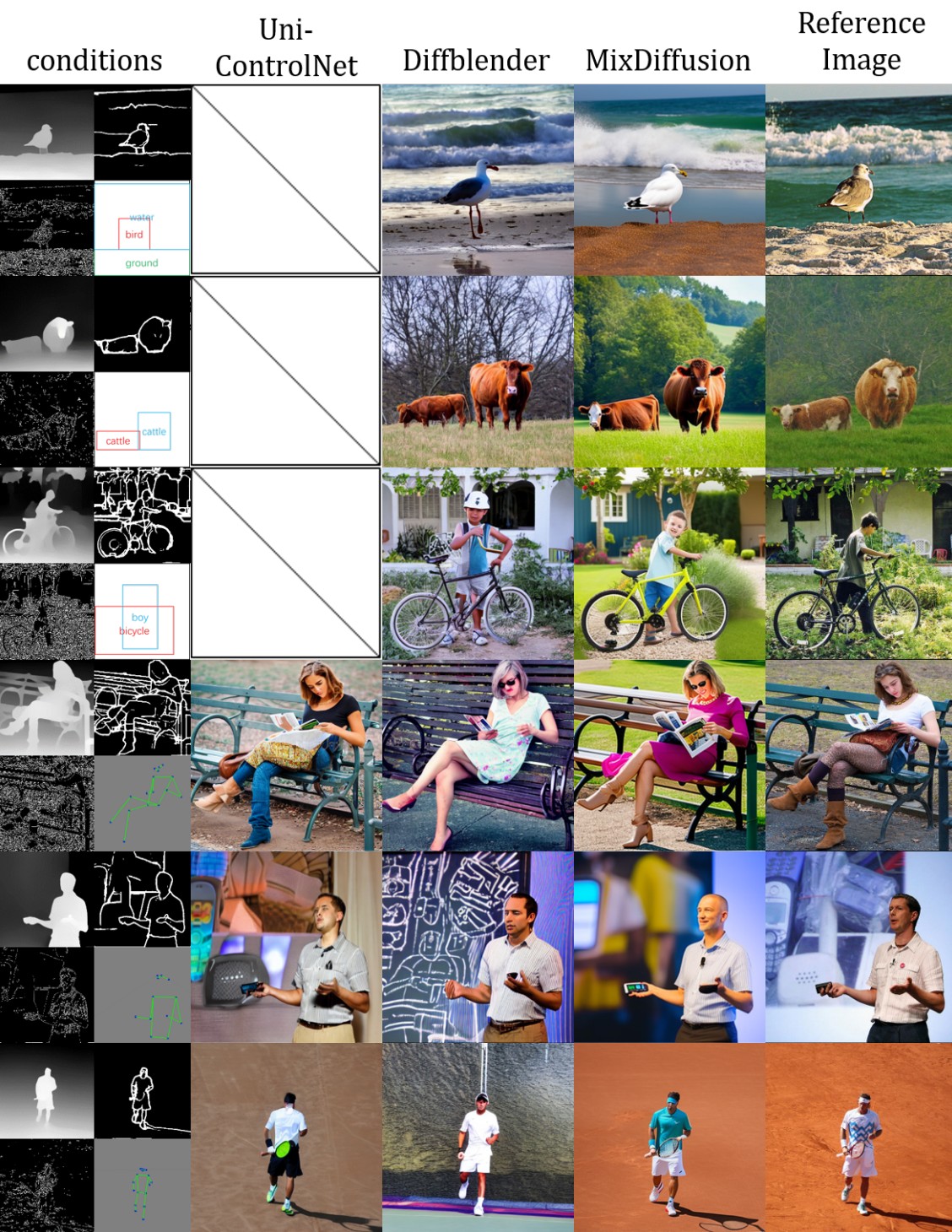

Figure 5: Comparison results of different controllable image generation models. Generation results of the Uni-ControlNet are not presented in the case of bounding box is included in the control condition inputs (i.e., the first three rows) since the Uni-ControlNet does not support bounding box input. Uni-ControlNet and DiffBlender perform worse than MixDiffusion in terms of background generation, consistency between generated images and conditions, and image detail quality.

## A.4 MixDiffusion Gallery

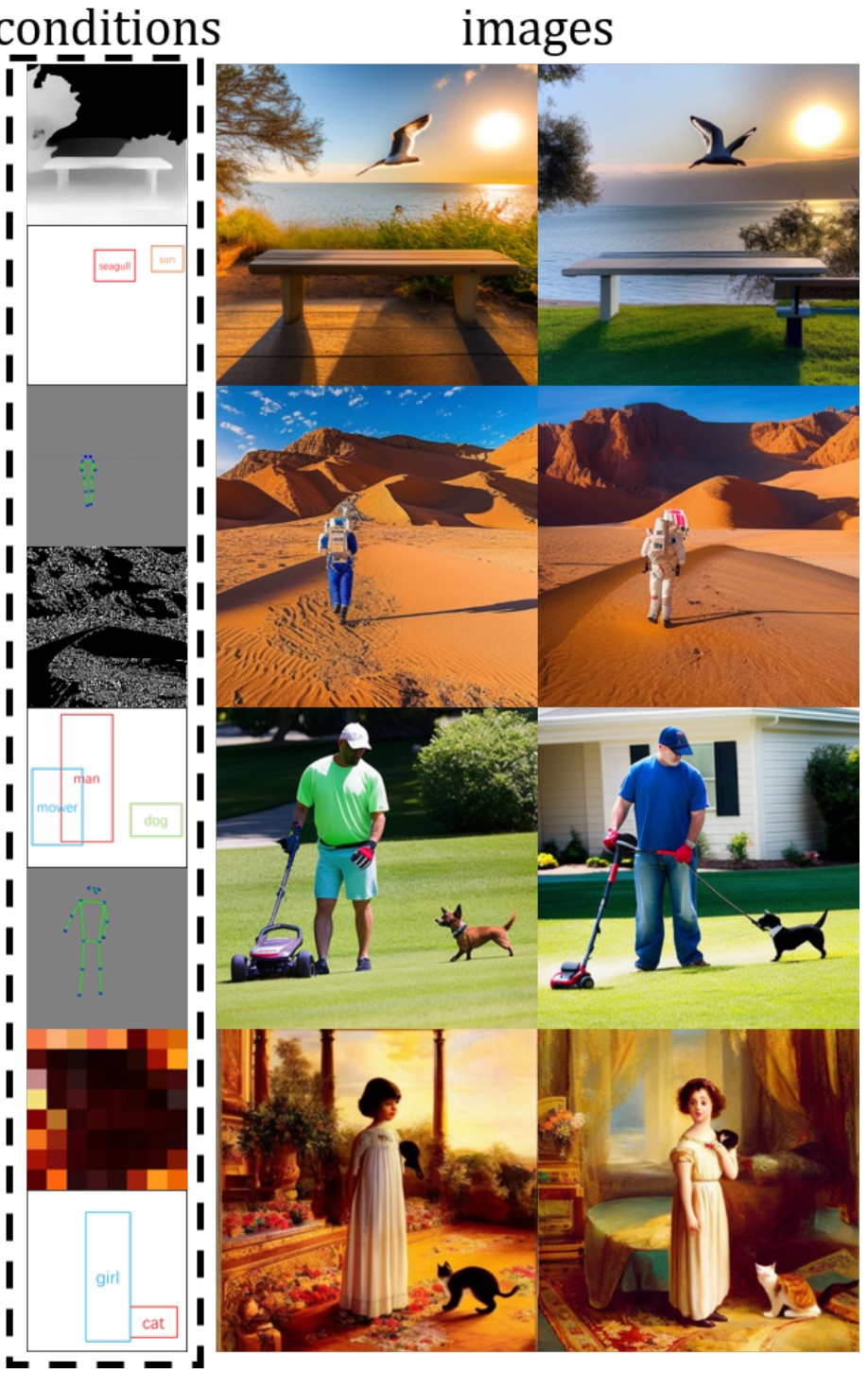

Figure 6: Images generated with our proposed MixDiffusion.

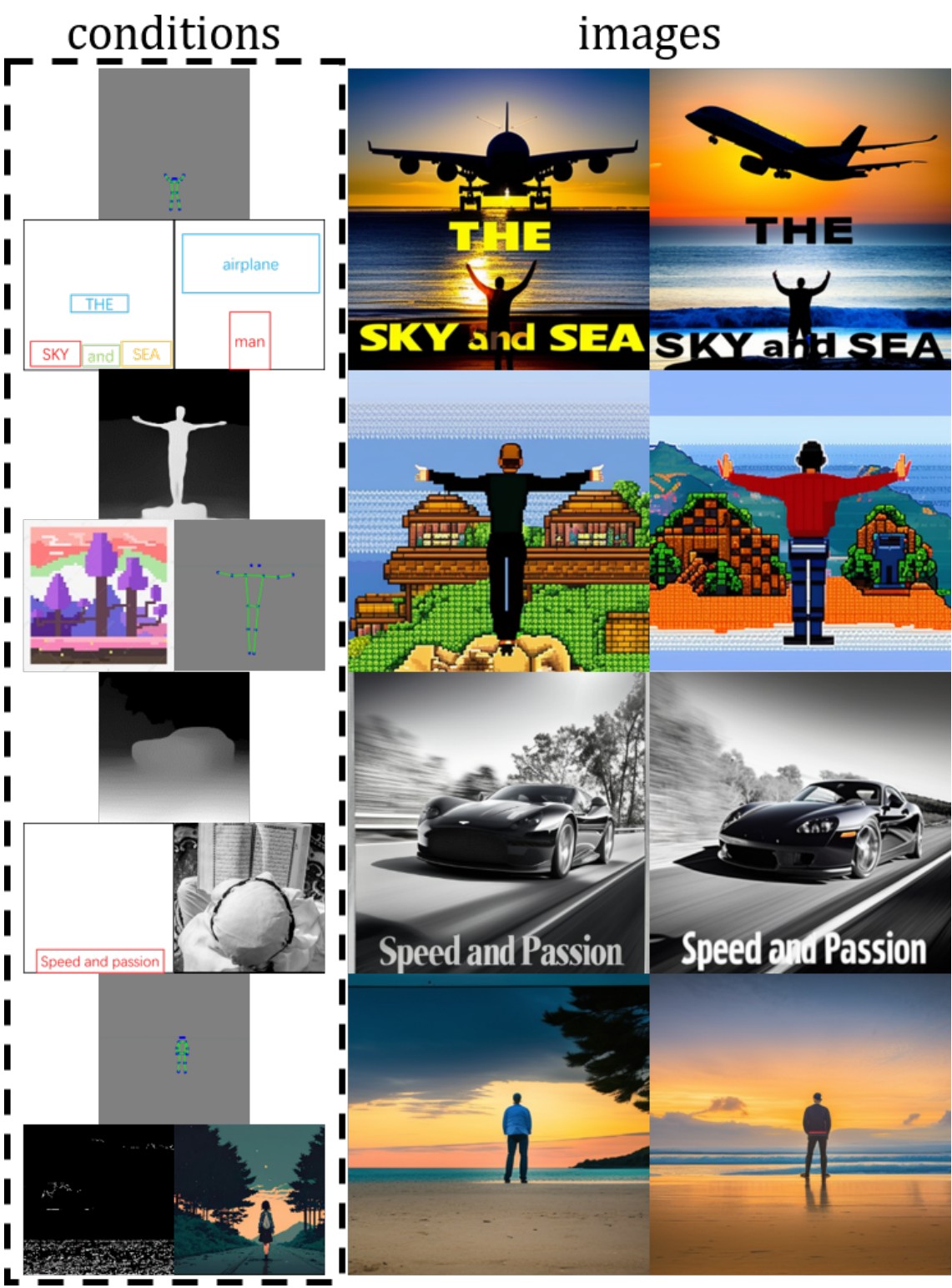

Figure 7: Images generated with our proposed MixDiffusion.

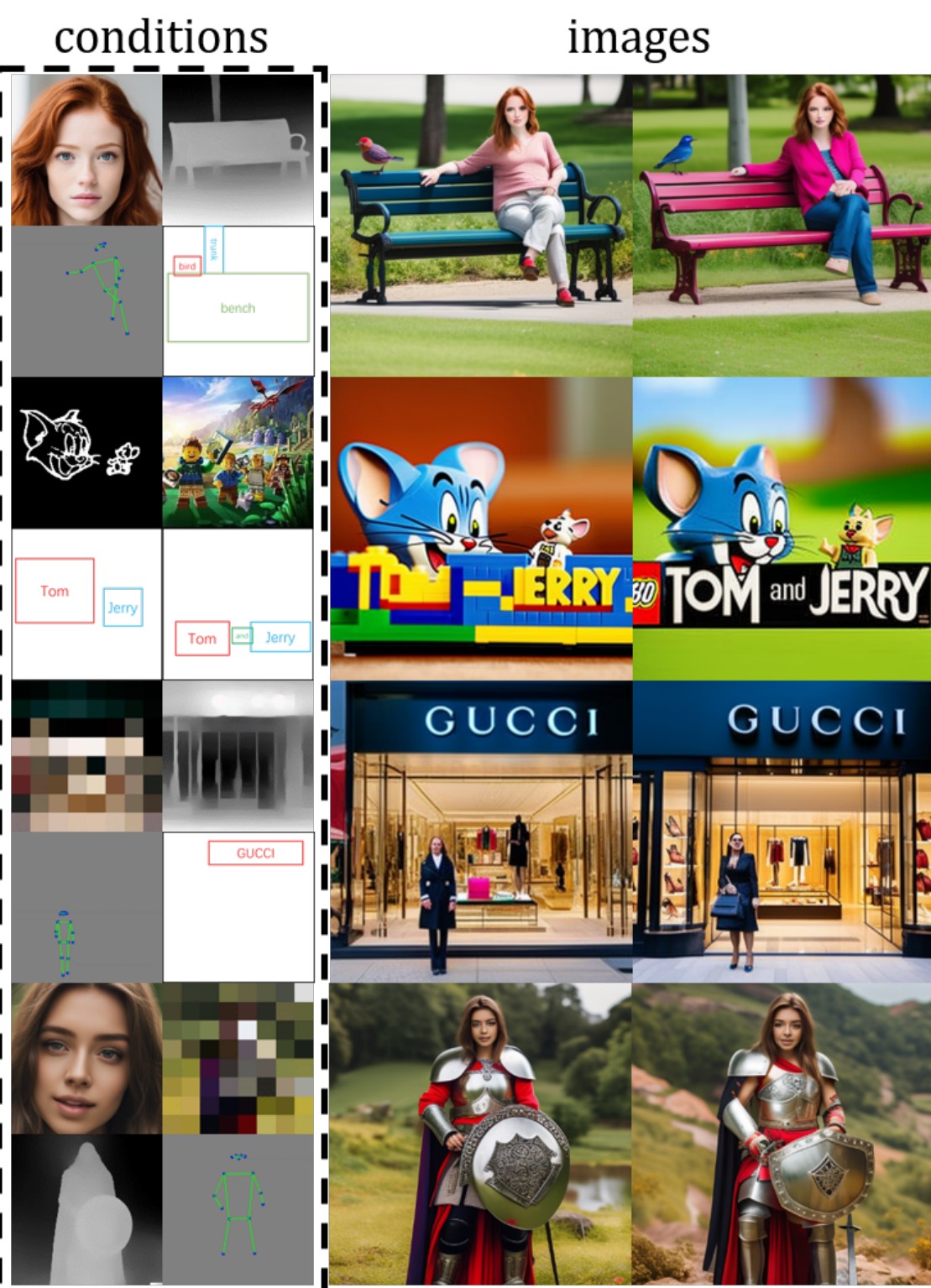

Figure 8: Images generated with our proposed MixDiffusion.

## A.5 THE USE OF LLMS

We used large language models (LLMs) to assist in polishing and refining the sentences in this paper. The content, ideas, and analyses are entirely our own, with LLMs employed solely for language enhancement purposes.

