# OpenReview forum: "MixDiffusion: Mixing Diffusion-based Uni-condition Text-to-Image Generation Models for Multi-conditions Image Synthesis"
_ICLR.cc/2026/Conference — ICLR 2026 Conference Withdrawn Submission_

### Official Review · Reviewer_5zKE · 2025-10-29

**Soundness:** 2
**Presentation:** 2
**Contribution:** 2
**Rating:** 2
**Confidence:** 5

**Summary:**

This paper presents MixDiffusion, a training-free framework that integrates multiple pre-trained uni-condition diffusion-based text-to-image models to enable flexible and unlimited multi-condition controllable image generation.
It bridges the gap between real-world applications that require multiple control conditions and current models that handle only one or two.
The authors further extend the theoretical foundation of diffusion models by deriving and proving an integration formula that combines the predicted noise distributions from individual uni-condition models to form a multi-condition model.
Extensive experiments demonstrate MixDiffusion’s flexibility and its superior performance in controllable, high-fidelity image generation.

**Strengths:**

- The paper introduces a training-free integration framework that combines multiple pre-trained diffusion models, providing a flexible and scalable solution for multi-condition image generation.
- It extends diffusion model theory by deriving and proving an integration formula that connects uni-condition and multi-condition noise distributions.
- Extensive experiments across various control conditions demonstrate superior image quality and condition alignment without additional training.

**Weaknesses:**

- ControlNet, T2I-Adapter, and IP-Adapter are plugin-based methods that support combining multiple conditions. The paper does not theoretically explain how MixDiffusion fundamentally differs in capability, scope, or limitations, nor does it provide direct comparisons under the same input conditions.

- Dense (Canny/Depth) and sparse (Pose/Box) controls influence noise differently. Simple linear fusion may let weaker signals be overridden, and no analysis or experiments are provided to verify robustness.

- MixDiffusion requires multiple Uni-condition models to run simultaneously, unlike some existing methods. The paper does not compare memory usage, inference speed, or computational complexity, limiting practical evaluation.

- Results mainly rely on quantitative metrics, with few qualitative visual comparisons. Supplemental examples mostly use multi-condition inputs derived from the same image, which does not reflect realistic combinations of controls from different sources.

- Integration Intensity shows metric improvements, but no generated samples are provided. Its effect on local details and consistency of control remains unverified.

**Questions:**

- How does MixDiffusion differ from plugin-based methods like ControlNet or T2I-Adapter, and can a direct comparison under the same inputs be shown?

- How does the model prevent weak sparse or dense signals from being overridden, and is there evidence of robustness?

- Can memory usage, inference speed, and complexity be compared with existing multi-condition methods?

- Could results using multi-condition inputs from different sources be provided to reflect real-world scenarios?

- Can visual examples show how Integration Intensity affects local details and control consistency?

---

### Official Review · Reviewer_LtXX · 2025-10-29

**Soundness:** 2
**Presentation:** 1
**Contribution:** 3
**Rating:** 4
**Confidence:** 4

**Summary:**

This paper proposes a training-free method that enables many conditional controls in diffusion-based image generation. The key idea is to integrate the predicted noise from multiple single-condition diffusion models, each trained for a different condition, thereby achieving multi-condition controllability. Moreover, the integration approach is theoretically derived and justified. The authors compare their method with both uni- and multi-condition models, showing superior image quality and controllability.

**Strengths:**

This paper's strengths are as follows.

(1) The proposed framework allows adding multiple control conditions without additional training. It introduces a theoretically grounded way of integrating predicted noises from different single-condition models to achieve multi-condition controllability. The paper provides valuable theoretical insight into noise integration within diffusion models.

(2) The method quantitatively demonstrates superior performance in terms of both controllability and image quality compared to conventional methods, despite being training-free.

**Weaknesses:**

This paper's weaknesses are as follows.

(1) The approach still requires specialized uni-condition models for each control type. It does not extend a single training-free T2I model to handle multiple conditions directly. Therefore, it may not be entirely cost-free, as several pre-trained models are needed. The notion of “training-free” becomes somewhat ambiguous. Previous studies, such as [1] and [2], have used this term for a single model that can operate without additional training.  In contrast, this paper’s approach relies on multiple independently trained models, which may reduce its novelty or practicality compared to other training-free frameworks.

(2) Equation (6), which represents the core of this work, is insufficiently explained. The derivation process should be presented more clearly. Moreover, there are three instances of ε_fit in the equation — it is unclear whether they are equivalent and which form is actually used in implementation.

(3) Evaluation metrics are inconsistently applied. Table 2 evaluates image quality, text alignment, and conditional controllability, while Tables 1 and 3 focus only on controllability. Since controllable generation requires assessment of both controllability and quality, the latter tables should also include metrics such as FID or ImageReward.

---
[1] Sicheng M, et al., "FreeControl: Training-Free Spatial Control of Any Text-to-Image Diffusion Model with Any Condition," CVPR2024
[2] Jiwen Yu, et al., "FreeDoM: Training-Free Energy-Guided Conditional Diffusion Model," ICCV2023

**Questions:**

My questions about this paper are as follows.

(1) The paper states that multiple models share a single Text Encoder and VAE.
 Is this feasible only when all uni-condition models are derived from the same pre-trained backbone (e.g., Stable Diffusion)? Please clarify this assumption.

(2) Please provide more explanation for Equation (4). On what basis is it assumed that the predicted noise follows a Gaussian distribution? This assumption does not appear explicitly in DDPM[1] or LDM[2] papers, so its theoretical justification is unclear.

(3) If the independence assumption between conditions breaks down during denoising, to what extent does the theoretical guarantee of MixDiffusion still hold?

(4) Please clearly specify which uni-condition model is used for each control type in MixDiffusion. A summary table (e.g., in the Appendix) would improve clarity.

---
[1]  Jonathan Ho, et al., "Denoising Diffusion Probabilistic Models," NeurIPS2020
[2] Robin Rombach, et al., "High-Resolution Image Synthesis with Latent Diffusion Models," CVPR2022

---

### Official Review · Reviewer_3rj3 · 2025-10-31

**Soundness:** 3
**Presentation:** 3
**Contribution:** 2
**Rating:** 2
**Confidence:** 5

**Summary:**

The paper proposes MixDiffusion, a training‑free framework that composes multiple pre‑trained uni‑condition text‑to‑image diffusion models (e.g., for boxes, pose, depth, canny, sketch, faces, visual text) into a single multi‑condition generator. The core idea is a product‑of‑experts–style integration applied at every denoising step: treat each uni‑condition model’s predicted noise as a Gaussian.

**Strengths:**

1. Simple, principled integration. The closed‑form update for the integrated noise (Eq. (6)) is elegant and easy to implement, with a clear probabilistic justification via Bayes and a Gaussian product‑of‑experts view.

2. Training‑free & modular. The framework composes off‑the‑shelf uni‑condition models and shares a single text encoder and VAE across them (Fig. 2), simplifying deployment and enabling rapid addition of new controls.

**Weaknesses:**

- **Incremental novelty of weighted guidance.** The paper’s main contribution—combining multiple uni-condition denoisers via a weighted guidance or product-of-experts rule (Eq. 6)—follows an already well-explored idea in multi-control diffusion literature[1-4]. The conceptual novelty is therefore limited.

- **Limited performance gain over baselines.** Although MixDiffusion achieves competitive OKS and IoU, the quantitative improvements over prior methods are small or inconsistent (see Tables 1–2). The paper does not include significance tests or detailed error analysis, so the claimed superiority is not convincingly supported.

- **High computational cost with increasing conditions.** When handling \(k\) control inputs, MixDiffusion requires \(k\) separate denoising models, leading to linear growth in memory and latency. This is less efficient than approaches that share one denoiser with multiple lightweight adapters.

- **Unverified independence assumption.** In Eq. (5), the method assumes that all control conditions are independent so that the joint likelihood factorizes. However, the paper provides no empirical justification or analysis for this strong assumption, despite its clear influence on theoretical validity.

[1] Z. Huang et al. Collaborative diffusion for multi-modal face generation and editing. In CVPR, pp. 6080–6090, 2023.

[2] Y. Wang et al. High-fidelity person-centric subject-to-image synthesis. In CVPR, pp. 7675–7684, 2024.

[3] L. Wang et al. Decompose and realign: Tackling condition misalignment in text-to-image diffusion models. In ECCV, pp. 21–37. Springer, 2024.

[4] P. Cao et al. Image is all you need to empower large-scale diffusion models for in-domain generation. In CVPR, pp. 18358–18368, 2025.

**Questions:**

See weaknesses 1&4

---

### Official Review · Reviewer_ehqz · 2025-11-01

**Soundness:** 3
**Presentation:** 3
**Contribution:** 2
**Rating:** 4
**Confidence:** 5

**Summary:**

The paper introduces MixDiffusion, a training-free method that enables multi-condition text-to-image generation by mixing multiple pre-trained uni-condition diffusion models (e.g., for pose, canny, depth, segmentation). The authors proposed an integration intensity strategy to gradually reduce the influence of control conditions during later denoising steps for improved visual coherence. Extensive experiments on COCO 2017 and MARIO Eval show that MixDiffusion achieves better controllability than baseline methods.

**Strengths:**

**Clear Writing:**  The proposed method is clearly presented, and the introduction is well-written. Most terms are properly explained, making the paper comfortable and easy to read.

**Effective Integration:** The proposed method enables joint utilization of multiple control modules (e.g., ControlNet, T2I-Adapter, and GLIGEN), allowing users to flexibly combine existing pre-trained diffusion models and potentially support an arbitrary number of control signals.

**Weaknesses:**

1. **High Inference Cost:** To handle N control maps, MixDiffusion requires N forward passes through N different control modules. Hosting these modules on the same device significantly increases GPU memory usage. Meanwhile, the performance improvement is relatively marginal compared to baseline methods—for example, MixDiffusion’s CLIP score (14.42) vs. T2I-Adapter’s 14.27, and its Depth RMSE (31.21) vs. T2I-Adapter’s 30.51.

2. **Limited Novelty and Practical Flexibility:** Previous training-free diffusion-based T2I frameworks—such as Plug-and-Play [1], Universal Guidance [2], Pix2Pix-Zero [3], and Diffusion Self-Guidance[4]—have demonstrated broader flexibility and unlocked new applications where real data is scarce. In contrast, MixDiffusion’s flexibility primarily lies in combining spatially controlled diffusion models and relies on strong assumptions that all models share the same latent space or sampler. When new control-map modalities are introduced, MixDiffusion still requires a separately trained model for that modality, limiting its overall adaptability.

3. **Incorrect Citation Format:** Many citations in the paper use an incorrect format. For example, in L389, it should be “ControlNet (Zhao et al., 2024), Diffblender (Kim et al., 2023)” rather than the current form.

4. **Missing Reference:** In L387, the authors claim that “there is no training-free multi-condition image generation model,” but FreeControl [5] (figure 14) already demonstrated the ability to process multiple condition maps in a training-free manner. A more thorough literature review is recommended to maintain good academic practice.


**Reference:**
1. Naret et al., Plug-and-Play Diffusion Features for Text-Driven Image-to-Image Translation, CVPR 2023, Link: https://arxiv.org/pdf/2211.12572
2. Arpit et al., Universal Guidance for Diffusion Models, ICLR 2024, Link: https://arxiv.org/abs/2302.07121
3. Gaurav et al., Zero-shot Image-to-Image Translation, SIGGRAPH 2023, Link: https://arxiv.org/abs/2302.03027
4. Dave et al., Diffusion Self-Guidance for Controllable Image Generation, NeurIPS 2023, Link: https://arxiv.org/abs/2306.00986
5. Sicheng et al., FreeControl: Training-Free Spatial Control of Any Text-to-Image Diffusion Model with Any Condition, CVPR 2024, Link: https://arxiv.org/abs/2312.07536

**Questions:**

How will MixDiffusion work on Flow-Matching-based models? Like FLUX or SD3.5

---

### Note · Authors · 2025-11-12

**Comment:**

We take the reviewer's suggestions into account and realize that the experiment needs improvement and the paper requires significant changes.

**Withdrawal Confirmation:**

I have read and agree with the venue's withdrawal policy on behalf of myself and my co-authors.